# Bistable Boron-Related Defect Associated with the Acceptor Removal Process in Irradiated *p*-Type Silicon—Electronic Properties of Configurational Transformations

**DOI:** 10.3390/s23125725

**Published:** 2023-06-19

**Authors:** Andrei Nitescu, Cristina Besleaga, George Alexandru Nemnes, Ioana Pintilie

**Affiliations:** 1National Institute of Materials Physics, Atomistilor 405A, 077125 Magurele, Ilfov, Romania; andrei.nitescu@infim.ro (A.N.); cristina.besleaga@infim.ro (C.B.); 2Faculty of Physics, University of Bucharest, 077125 Magurele, Ilfov, Romania; nemnes@solid.fizica.unibuc.ro; 3Horia Hulubei National Institute for Physics and Nuclear Engineering, 077126 Magurele, Ilfov, Romania

**Keywords:** radiation hard detectors, boron-doped irradiated silicon, bistable defect, acceptor removal, configurational transformations, LGAD, TSC technique

## Abstract

The acceptor removal process is the most detrimental effect encountered in irradiated boron-doped silicon. This process is caused by a radiation-induced boron-containing donor (BCD) defect with bistable properties that are reflected in the electrical measurements performed in usual ambient laboratory conditions. In this work, the electronic properties of the BCD defect in its two different configurations (A and B) and the kinetics behind transformations are determined from the variations in the capacitance-voltage characteristics in the 243–308 K temperature range. The changes in the depletion voltage are consistent with the variations in the BCD defect concentration in the A configuration, as measured with the thermally stimulated current technique. The A→B transformation takes place in non-equilibrium conditions when free carriers in excess are injected into the device. B→A reverse transformation occurs when the non-equilibrium free carriers are removed. Energy barriers of 0.36 eV and 0.94 eV are determined for the A→B and B→A configurational transformations, respectively. The determined transformation rates indicate that the defect conversions are accompanied by electron capture for the A→B conversion and by electron emission for the B→A transformation. A configuration coordinate diagram of the BCD defect transformations is proposed.

## 1. Introduction

Silicon detectors are intensively used for fundamental research in high-energy physics (HEP) experiments, research with photons or radiation in free electron lasers, space missions, fusion experiments, as well as medical applications [1,2,3,4,5]. The main reasons for the extended use of these detectors are the flexibility of their structural design for extreme high spatial and time resolution with a high signal-to-noise ratio, the possibility for electronic integration on the same chip, and a large amount of experience in semiconductor process technology. Among the various applications of silicon sensors, the most demanding one is their operation in the increased intensity of radiation fields required in particle physics experiments. The impact of energetic particles on sensor material leads to displacement damage effects; the impinging particles having enough energy to dislocate the atoms from their lattice site and further form electrically active defects, which degrade the detector’s performance, ultimately limiting the sensors’ practical use by increasing the radiation intensity [6]. The radiation damage effects, occurring in the present large hadron collider (LHC) experiments at CERN, can be tolerated by the installed *n-type silicon* sensors for an overall operational period of 10 years. These sensors were designed to withstand an integrated luminosity of 300 fb^−1^ corresponding to a cumulated radiation level of 2 × 10^15^ n_eq_/cm^2^ (n_eq_ stands for 1 MeV neutron equivalent) and ionizing doses of about 300 kGy. Common to *n-type* silicon sensors is *the type inversion*, a phenomenon occurring gradually with an increasing irradiation fluence [7,8] and not as a result of plastic deformations induced at elevated temperatures [9]. Thermally stimulated current (TSC) experiments evidenced that the space charge sign inversion in irradiated n-type silicon diodes is determined by the generation of some specific acceptor defects [10,11,12]. Such defects cause a switch of the high electric field from the structured readout side to the backside of the detector, leading to a loss in spatial resolution and a reduced charge collection efficiency. Therefore, for the high-luminosity LHC (HL-LHC) with an integrated luminosity of 4000 fb^−1^ (cumulated irradiation fluence of 3 × 10^16^ n_eq_/cm^2^) [13], upgrade foreseen for 2027, new generations of tracking and timing detectors have to be developed. A solution proposed and tested by the RD50 community [14,15] was to develop devices based on *p-type* silicon, which do not invert, from simple n^+^-p silicon diodes to devices amplifying signals such as low gain avalanche detectors—LGADs [16,17,18,19,20] and CMOS monolithic active pixel sensors [21,22,23,24,25,26,27,28,29]. A specific detrimental effect observed in all types of *p-type* silicon sensors is the loss of initial acceptor doping during irradiation at ambient temperatures, a process known as acceptor removal (AR) [16,17,18,19,20,21,22,23,24,25,26,27,28,29,30,31,32]. The AR process is primarily observed from the changes induced by irradiation in the capacitance-voltage (*C-V*)/current-voltage (*I-V*) characteristics of the devices. A decrease in the depletion voltage (*V_dep_*) with increasing irradiation levels is observed. Consequently, a reduction in the effective space charge density (*N_eff_*) takes place as well. While most of the experimental studies evidencing the AR phenomenon are performed on boron-doped silicon, the effect was also observed in Ga-doped *p-type* silicon [20]. At the microscopic level, the AR process was studied on boron-doped silicon and associated with the formation of a boron-containing donor (BCD), with an energy level showing a field-enhanced emission (Poole Frenkel effect) [33]. This energy level was determined to be located between 0.245 eV and 0.28 eV from the conduction band (*E_c_*) of silicon depending on the value of the electric field in the investigated device [32,34,35,36,37,38,39,40,41,42]. It has been shown that for low and intermediate doping levels, the introduction rate of the BCD defect increases linearly with the boron and oxygen content in the material, while a competing defect reaction, leading to the production of a C_i_O_i_ (interstitial carbon–interstitial oxygen) complex, is revealed [35]. The positively charged state of the BCD defect with an energy level close to E_c_ and the competing reaction involving interstitials indicates that this radiation-induced complex is of interstitial type [34]. In the 150–200 °C temperature range, the BCD defect dissociates, and substitutional boron (B_S_) is recovered [34,35,36,40,41,42]. It is worth mentioning that B_S_ can also react with silicon vacancy (V) or divacancy (V_2_) and form B_S_V and B_S_V_2_ complexes [43,44,45,46]. These vacancy-type defects have different characteristics compared to those of BCD in terms of generation, thermal stability, and charge trapping. The B_S_V defect dissociates at 260 K, and thus it is not detected in samples irradiated at ambient temperatures [43,44]. On the other hand, B_S_V_2_ can form only in heavily doped material [44,45] or after annealing out divacancies at temperatures above 250 °C [46]. Thus, the AR process in samples irradiated at ambient temperatures is explained by considering a reaction between the substitutional boron (B_s_) and silicon interstitials (Si_i_) created by irradiation. The traditional view of explaining the deactivation of the boron acceptor dopant and the formation of a donor instead accounts for two steps: (first) substitutional boron (B_s_) atoms switch places with the interstitial Si via the Watkins replacement mechanism, becoming interstitials (B_i_), losing the acceptor character; (second) B_i_ atoms migrate in the crystal, combine with abundant oxygen interstitial centers (O_i_), and form B_i_O_i_ defects which have a donor energy level in the bandgap of silicon [36,42]. Another mechanism, proposed more recently for all types of acceptor dopants in silicon, not implying the formation of interstitial dopant atoms and their migration at ambient temperatures, accounts for the reaction between the negatively charged acceptor dopant (A_Si_) and the positively charged Si_i_ atom created by irradiation [47,48]. This way, the A_Si_-Si_i_ complex is created, a defect for which three charge states (−,0,+) and three configurations are possible. For boron-doped silicon, this reaction path leads to the formation of the B_s_Si_i_ defect. Such an AR mechanism becomes likely according to secondary ion mass spectroscopy (SIMS) measurements performed on boron-doped LGAD structures before and after irradiation where, after irradiation with several particles and fluences, the migration of boron atoms could not be detected [49,50]. According to the literature, both types of boron-containing defects, B_i_O_i_ and B_s_Si_i_, have donor energy levels [44,47,48] in the bandgap of silicon and, thus, when they are in the positive charge state, contribute twice their concentration to the reduction in the initial *p-type* doping of silicon. As the chemical structure of these defects is under debate, we will further refer to them as BCD. Recently, the metastability of BCD complexes and their effect on the macroscopic properties of the devices was clearly evidenced in high-resistivity boron-doped silicon diodes irradiated with high fluences of 1 MeV neutrons [32]. It has been shown that the radiation-induced BCD defect is bistable, being able to reversibly switch between two structural configurations: A—the electrically active one associated with the detected donor state of the defect (*BCD_A_*^(0/+)^), with its specific energy level located around 0.25 eV from the conduction band of silicon, and B—a neutral-charged one (*BCD_B_*^0^) observed indirectly in electrical measurements by following the loss in the TSC signal deriving from the *BCD_A_*^(0/+)^ state. The change in the configuration from A to B can be triggered by an excess of carriers at ambient temperatures, achieved even unintentionally by the inherent short exposure to ambient light when manipulating the samples for starting the desired experiments. The relaxation from the B to A ground state takes place slowly if the sample is kept in the dark. Both configurational conversions cause changes in the *C-V/I-V* characteristics and, in the determined value of *N_eff_*, the main parameter considered in the evaluation of the AR process. The time scales of fully switching between the two configurations in high-resistivity diodes are large enough to be properly characterized. The goal of the present study is to determine the electronic properties of the BCD in its different configurations (energy levels) and the kinetics of configurational conversions in conditions similar to those usually encountered in laboratories.

## 2. Materials and Methods

A pair of high-resistivity samples, consisting of a 45 µm thick n^++^-p-p^++^ pad diode and a n^++^-p^+^-p-p^++^ LGAD structure, both produced on 4″ silicon-on-insulator (SOI) wafers on a 300 µm thick support wafer and 1 µm buried oxide by *Centro Nacional de Microelectrónica, Barcelona, Spain* (CNM), have been investigated in this study. The front and back areas of the electrodes are *A_f_* = 3.3 × 3.3 mm^2^ and *A_b_* = 5 × 5 mm^2^. These samples are labelled further on as PAD and LGAD. The gain layer (p^+^) of the LGAD sample is about 1 µm thick and was implanted with boron in a concentration of 4 × 10^16^ cm^−3^ in the 45 µm thick pad diode. Both diodes have guardrings, which are grounded in all the electrical measurements, and small circular open windows in the front contact allow illumination—see Figure 1. 

The guardrings need to be grounded when bulk damage effects are investigated. This way, the contribution of surface defect states on the electrical characteristics of the devices are diminished, and an accurate determination of defect concentrations in TSC experiments can be achieved [10,11,12,51]. In fully or over-depleted diodes, the active area of the diodes is always defined by the front electrode area (*A_f_*) no matter whether space charge sign inversion occurs or not in the bulk of the samples. 

The samples were irradiated with 1 MeV neutrons at the Triga Reactor in Ljubljana, with a fluence of 10^14^ n/cm^2^. For the present study, the samples were annealed for 136.000 min at 80 °C. After such long annealing, all the radiation-induced defects are thermally stable. On these samples, we performed dedicated experiments to characterize the change in the configuration of the BCD defect and the subsequent variation in *N_eff_*. Variations in the concentration of the *BCD_A_*^(0/+)^ defect’s configuration are evidenced by TSC experiments which are performed and analyzed according to the procedures and mathematical formalism described in [51,52]. The *N_eff_* values, for different changes in the BCD configuration, are determined from the corresponding *C-V* characteristics using the following relation:(1)$$\frac{d(\frac{1}{C^{2}})}{dV}=-\frac{2}{A^{2}\varepsilon \epsilon_{0}q_{0}\left|N_{eff}\right|}$$

The change in the BCD configuration was previously evidenced by exposing similar *p-type* PAD diodes for a short time to ambient light or subjecting them to a thermal treatment at 80 °C, conditions that are usually encountered in typical radiation damage studies [32]. It has been shown that, after only 15 min of exposure of the diodes to ambiental laboratory light, *N_eff_* significantly changes, and it takes more than 7 h at ambient temperature to relax to its stable value. Any measurement performed in between will result in a different value of *N_eff_*. Similar behavior has been observed in the *BCD_A_*^(0/+)^ defect concentration, determined from TSC measurements. Thus, it has been concluded that the bistable character of the BCD defect is causing such long-time variations in *N_eff_* at ambient temperatures. For the present study, where transition rates and energy barriers for switching between the BCD configurations are to be determined, precise measurement procedures at constant temperatures have to be employed. Thus, while the TSC technique can provide quantitative information about the defect concentration in the A configuration, the results obtained during the TSC temperature scan cannot be used to determine the kinetics of configurational transformations. Instead, we used the *C-V* characteristics measured at different temperatures to determine the transition rates and energy barriers corresponding to the BCD switch from A to B configuration and back. While the change from A to B configuration takes place in non-equilibrium conditions, the reverse process occurs in the absence of free-carriers in excess [32]. For following the A to B transformation and to be sure that the same excess of carriers is induced in the whole volume of the samples, we injected a small forward current of 5.7 µA, similar to the short-circuit photocurrent generated by laboratory ambient light. Then, *C-V* characteristics at different temperatures were measured after different injection times were performed in between the measurements. For following the B to A transformation, the BCD defect was first brought in its B configuration by a long enough time injection of 5.7 µA. Then, the injection was stopped, and several *C-V* measurements were taken over time. In between the measurements, the samples were kept in the dark and under 0 V. A 10 kHz frequency is used in all the *C-V* measurements. TSC experiments were employed for determining the concentration of BCD defects in the A configuration—[*BCD_A_*^(0/+)^]. In all the TSC measurements, the samples were cooled to *T*_0_ = 10 K where filling of the BCD defects was achieved by a forward current injection of 350 µA. Then, the samples were reverse biased and heated with a constant rate of *β* = 11 K/min. The applied reverse bias (*V_R_*) during heating was large enough to ensure the full depletion of the samples over the entire TSC temperature scan. In such a case, the TSC signal generated by a density of *N_t_* homogeneously distributed electron traps is given by [51,52,53]: (2)$${TSC}^{e}\left(T\right)=\frac{1}{2}\times q_{0}\times A_{f}\times d\times e_{n}(T)\times n_{t}(T_{0})\times \exp(-\frac{1}{\beta }\int_{T_{0}}^{T}\left(e_{n}\left(T^{\prime }\right)+e_{p}\left(T^{\prime }\right))dT^{\prime }\right)$$
where *q*_0_ is the elementary charge and *d* is the thickness of the device. 

The emission rates *e_n,p_*, as defined by Shockley–Read–Hall statistics, are [54,55]:(3)$$e_{n}\left(T\right)=c_{n}\left(T\right)\times N_{c}\left(T\right)\times \exp\left(-\frac{E_{C}-E_{t}}{k_{B}\times T}\right)$$
(4)$$e_{p}\left(T\right)=c_{p}\left(T\right)\times N_{v}\left(T\right)\times \exp\left(-\frac{E_{t}-E_{V}}{k_{B}\times T}\right)$$
where *N_C,V_* are the density of the states in the conduction/valence band, *v_th,n,p_* are the thermal velocities for electrons/holes [56], *k_B_* is the Boltzmann constant, *E_t_* is the defect energy level, and *c_n,p_* are the capture coefficients for electrons/holes defined as:(5)$$c_{n,p}\left(T\right)=\sigma_{n,p}\left(T\right)\times v_{th,n,p}\left(T\right)$$
where *σ_n,p_* are the defect capture cross-sections for electrons (*n*) and for holes (*p*). 

The *n_t_*(*T*_0_) factor in Equation (2) represents the amount of the traps filled with electrons during the high-level injection performed at *T*_0_ = 10 K. By solving the Shockley–Read–Hall equations [54,55] for the stationary case of high-level bipolar injection [57] at *T*_0_, it results that *n_t_(T*_0_*)* depends on the values of *c_n_* and *c_p_* according to the relation:(6)$$n_{t}\left(T_{0}\right)=N_{t}\times \frac{c_{n}(T_{0})}{c_{n}\left(T_{0}\right)+c_{p}(T_{0})}$$

Direct capture cross-section measurements for the BCD defects reveal values of 1.05 × 10^−14^ cm^2^ and 2.5 × 10^−20^ cm^2^ for *σ_n_* and *σ_p_*, respectively, both temperature-independent [32]. Considering that *σ_p_* << *σ_n_*, *c_p_* and *e_p_* can be neglected in Equations (2) and (6). However, because the BCD defect is a coulombic center, the electron emission rate becomes dependent on the applied electric field according to the 3D Poole Frenkel effect [33]:(7)$$e_{n}^{PF}\left(T\right)=e_{n,0}\left(\frac{1}{\gamma^{2}}\right)\left(e^{\gamma }\left(\gamma -1\right)+1\right)+\frac{1}{2}$$
with $\gamma =\frac{q_{0}}{k_{B}\times T}\sqrt{\frac{q_{0}\times \left|\vec{F}\right|}{\pi \times \varepsilon_{0}\varepsilon_{r}}}$, where *ε*_0_*ε_r_* is the dielectric constant of silicon and $\left|\overset{⃑}{F}\right|$ stands for a uniform electric field in the material [33]. The *e_n,_*_0_ term in Equation (7) is the emission rate in the absence of an electric field and is expressed by Equation (3), in which *E_t_ = E_t,_*_0_ represents the so-called zero-field activation energy. However, a position-dependent electric field has to be accounted for due to the diodes [11]. The local electric field for an applied reverse bias larger than depletion voltage (*V_R_* > *V_dep_*) is given by:(8)$$F\left(x,T\right)=\frac{q_{0}}{\varepsilon_{0}\varepsilon_{r}}\times N_{eff}\left(T\right)\times \left(d-x\right)+\frac{V_{R}-V_{dep}(T)}{d}$$
where *x* is the distance from the n^++^ side across the diode thickness while *V_dep_(T)* and *N_eff_(T)* are the full depletion voltage and the effective space charge density in the temperature range where the defect emits electrons, respectively. Considering the built-in potential of the diode *(V_bi_)*, the *V_dep_(T)* and *N_eff_(T)* values are connected via the relation:(9)$$V_{dep}\left(T\right)=\frac{q_{0}}{{2\times \varepsilon }_{0}\varepsilon_{r}}\times d^{2}\times \left|N_{eff}\left(T\right)\right|-V_{bi}(T)$$

With these considerations, Equation (2) becomes:(10)$${TSC}^{e}\left(T\right)=\frac{1}{2}\times q_{0}\times A_{f}\times N_{t}\times \int_{0}^{d}\left\{e_{n}^{PF}(x,T)\times \exp(-\frac{1}{\beta }\int_{T_{0}}^{T}e_{n}^{PF}(x,T)dT\right\}dx$$

Equation (10) describes the shape and magnitude of the TSC peak generated by the emission of electrons from a donor, and it can be used to numerically fit the measured TSC signal. This procedure allows not only the concentration (*N_t_*) but also the zero-field activation energy (*E_t,_*_0_) of the defect to be determined. 

## 3. Results

In the A configuration, the BCD defect has a donor energy level at about 0.25 eV from the conduction band of silicon and gives rise to a specific peak in TSC measurements in the 90–110 K temperature range. The defect is detected only in irradiated samples. As a donor in the upper part of the bandgap, the *BCD_A_*^(0/+)^ contributes at ambient temperatures with positive space charge in its full concentration. The defect can change from *BCD_A_*^(0/+)^ to *BCD_B_*^(0)^ configuration by capturing an electron and surmounting the energy barrier for configurational transformation E_A→B_. This way, the defect loses its donor activity and no longer contributes to the space charge in the diodes. Consequently, differences in the *C-V/I-V* characteristics at ambient temperatures and in the TSC peak corresponding to the *BCD_A_*^(0/+)^ configuration are encountered when the defect changes the configuration. In the present study, we used a small forward current injection of I_Fw_ = 5.7 µA for changing the BCD defect configuration from A to B. The *C-V* characteristics measured on PAD diodes after different durations of forward current injection at 293 K are shown in Figure 2a. The *C-V* curves shift towards larger biases and stabilize after approximately 3 h of injection. The increase in *V_dep_* is about 6.3 V. This means that *N_eff_* is increasing with about 5 × 10^12^ cm^−3^ due to the injection. Further on, by stopping the injection and monitoring the change in the *C-V* characteristics in time, a reverse effect is observed, and *V_dep_* is slowly decreasing back to the initial value before the injection has started in about 16 h—see Figure 2b. Similar behavior is observed for the high-resistivity *p-type* bulk of the irradiated LGAD diodes. The given time values represent the cumulated times passed in between the *C-V* measurements, of the I_Fw_ injection (Figure 2a), and after the end of the injection when the samples are kept under 0 V in the dark (Figure 2b). In non-irradiated samples, no variations in *C-V/I-V* characteristics are observed.

In general, such changes in *V_dep_*, and so in *N_eff_*, can be caused by variations in the concentration of acceptors, of donors, or of both. TSC investigations can clarify this aspect by showing exactly what type of defects encounter changes in the concentration. In Figure 3 are the TSC spectra measured on PAD and LGAD diodes after the small forward current injection was performed at 293 K for 3 h and the BCD defect switched from A to B configuration. However, the first TSC spectrum can be recorded only 2 h after the injection is performed. Similar to what was previously shown in [32], significant changes in the TSC peaks are observed only for the BCD defect, and these were detected only in the A configuration. 

One can observe that during successive TSC measurements, the *BCD_A_*^0/+^ peak is strongly diminished in the first TSC measured spectra and raises in the next ones, reaching a maximum after almost a day. The maximum variation in *BCD_A_*^0/+^ evaluated on PAD diodes from these measurements is Δ[*BCD_A_*^0/+^] = 3.8 × 10^12^ cm^−3^. The first (2 h) and the last (24 h) TSC measurements shown in Figure 3a were analyzed according to the mathematical formalism described in Section 2. The simulation results are given in Figure 4. 

In order to account for the spatial distribution of the electric field in the simulation of the *BCD_A_*^0/+^ peak (see Equations (7)–(10)), we used a *V_dep_* of 92 V. This value corresponds to the smallest V_R_ that has to be applied in the *BCD_A_*^0/+^ peak temperature range for obtaining the largest *BCD_A_*^0/+^ signal, and this was determined from TSC experiments with different applied reverse voltages after the BCD defect stabilizes in the A configuration (not shown here). For the *BCD_A_*^0/+^ state, a zero-field activation energy of *E_t,_*_0_ = 0.286 eV was obtained for the TSC simulations shown in Figure 4a,b. The concentrations of emitted electrons from the *BCD_A_*^0/+^ state in the two situations presented in Figure 4a and 4b are 2.5 × 10^12^ cm^−3^ and 6.3 × 10^12^ cm^−3^, respectively.

Considering that there is a 2 h delay between the 5.7 µA injection at 293 K and the recording of the first TSC spectrum, during which the BCD defect can partly switch back in the A configuration, the determined Δ[*BCD_A_*^0/+^] value can be associated with the variations in *N_eff_* determined from *C*-*V* characteristics at 293 K. The large difference between the magnitude of the TSC peaks in PAD and LGAD diodes is due to the charge multiplication effect in the gain layer of LGAD structures, which amplifies the electrons signal. Thus, the TSC signals associated with the emission of electrons is multiplicated, the amplification factor being above 10 for all the detected electron traps, which is in agreement with the gain measured by Lange et al. on similar LGAD samples [58]. 

It is worth mentioning that the TSC measurements on non-irradiated samples did not reveal any peaks. Therefore, all the TSC signals in Figure 3 correspond to radiation-induced defects. In addition, among the different detected traps, only the *BCD_A_*^0/+^ donor level has unstable behavior and can be associated with the variations in *N_eff_* at an ambient temperature. Thus, during the small forward current injection at 293 K, the BCD switches from A to B configuration, losing its donor character and causing an increase in *V_dep_* (see Figure 2a) and so in *N_eff_*. By removing the injection, the BCD switches back slowly from B to A configuration, regaining its donor character and thus causing a reduction in *V_dep_* (see Figure 2b) and *N_eff_*. Because these configurational conversions are reversible, BCD is a bistable defect. Such behavior can experimentally be characterized by determining the electronic properties of the defect in the two configurations and the transformation kinetics. For first-order kinetics, a defect transforms independently of other defects or impurities in the material, and its concentration *N* is given by:(11)$$N(t)=N_{0}*\exp\left(-k*t\right)$$
where *k* is the transformation rate and *N*_0_ is the initial defect concentration. The transformation rate has usually thermally activated behavior, as described by:(12)$$k=k_{0}\exp\left(-\frac{E_{a}}{k_{B}T}\right)$$
where the value of the pre-exponential factor *k*_0_, also known as the frequency factor, hints to the physical mechanism behind the defect transformation. Values in the order of 10^7^ s^−1^ indicate the capture of free carrier by multiphonon emission, while those in the order of 10^12^ s^−1^ point to the emission of free carriers [59,60]. 

Further, we consider that the variations in *N_eff_* are entirely due to the change in the structural configuration of the BCD defects. Thus, to determine the transformation rates and energy barriers, we performed *C-V* measurements on the PAD sample at different temperatures. The results are presented in the following subsections for each of the configuration changes. 

### 3.1. Transformation from A to B Configuration

The change from A to B configuration of the BCD defect can be well investigated following the effect on *N_eff_* of a small forward current injection in the 243–303 K temperature range. The variations in the *C-V* characteristics and *N_eff_* following the injection at 253 K and 303 K are given in Figure 5 and Figure 6, respectively. As reference for representing *ΔN_eff_*_,_ we considered the *N_eff_^0^* value determined from *C-V* curves prior to injection (0 min. in Figure 5a) when the sample is relaxed and the defect is in its *BCD_A_*^(0/+)^ ground state. Presuming that the increase in Δ*N_eff_* is caused entirely by the decrease in *BCD_A_*^(0/+)^ concentration via first-order kinetics, the Δ*N_eff_*(*t*) = *N_eff_*(*t*) − *N_eff_*^0^ can be described by:(13)$$\Delta N_{eff}\left(t\right)={-∆\left[{BCD}_{A}\left(t\right)\right]={BCD}_{A}}^{0}*(1-\exp\left(-k*t)\right)$$

The temperature dependence of the transformation rates *k* for A to B defect conversion can be determined by performing similar experiments and evaluations at different temperatures. The results obtained in the 243 K–303 K temperature range are given in the Arrhenius plot depicted in Figure 7. From this plot, and applying Equation (12) for the transformation rate, a value of *E_A__→B_* = 0.363 eV is determined for the energy needed to convert the BCD defect from A to B configuration. The pre-exponential factor for this transformation (*k*_0_*^A^^→B^*) is in the order of 10^3^ s^−1^ and it is much smaller than the value predicted for a free-carrier capture, an aspect that will be discussed in Section 4.

### 3.2. Transformation from B to A Configuration

In the absence of free electrons, the BCD defect slowly returns to its ground state A, recovering the donor character by emitting an electron. The conversion from B to A configuration determines a variation in *N_eff_* (see Figure 2b) which can be expressed as: (14)$$\Delta N_{eff}(t)={∆\left[{BCD}_{B}(t)\right]={BCD}_{B}}^{0}*(\exp\left(-k*t\right)-1)$$

For studying this process, we first brought the defect in its B configuration by applying the small I_Fw_ = 5.7 µA injection. Then, after stopping the injection, *C-V* measurements at different times of storing the sample under 0 V in the dark were performed. The conversion from B to A configuration is not observed at temperatures below 273 K, and even at this temperature, only 10^%^ of *N_eff_* is recovered in 3 days. Only from 283 K does the conversion from B to A take place in a reasonable time. In Figure 8 and Figure 9 are the variations in *N_eff_* in time for 283 K and 303 K, respectively. 

The variation in terms of temperature in the B to A configurational transformation rate is represented in the Arrhenius plot in Figure 10. For the B to A defect conversion, an energy barrier of *E_B__→A_* = 0.94 eV and a constant rate in the order of 10^12^ s^−1^ are determined. The *k*_0_*^B^^→A^* value is typical for free-carrier emission. 

## 4. Discussion

As mentioned previously, there is no clear identification of the BCD defect, and the physical origin of its bistability is unknown. We can, however, discuss the phenomenology behind the BCD conversion, from A to B configuration and back, with the help of a configuration coordinate (CC) diagram based on the energy barriers determined in our study. The CC diagram is depicted in Figure 11. 

Bistability refers to the cases when a defect can exist in two structural configurations for the same charge state. Examples of identified radiation-induced bistable centers in silicon can be found in references [60,61,62,63,64] and the cited literature. In the BCD case, the defect is positively charged at ambient temperatures when it exists in configuration A and is neutral when it converts to configuration B. This means that, for configurational conversion, the defect found in the *BCD_A_^+^* charge state has to first capture an electron and then to surmount the energy barrier *E_A__→B_* = 0.363 eV (see Figure 11a). The A→B reaction can be written as:(15)$${BCD}_{A}^{+}+e\to {BCD}_{B}^{0}$$

With the corresponding transformation rate of:(16)$$k=10^{3}\exp\left(-\frac{0.363\;\mathrm{eV}}{k_{B}T}\right)$$

The 10^3^ s^−1^ pre-exponential factor is much lower than that of 10^7^ s^−1^ usually expected for a thermally activated capture of an electron and can be explained by the effect of the Fermi level on the population of the donor energy level. Thus, Fermi energy laying below the defect energy level leads to a substantial decrease in the pre-exponential factor. Such a situation was discussed by *Chantre* [60] for bistable thermal donors in n-type silicon. In our case, the investigated diodes are *p-type*, and the Fermi level lies always in the lower part of the bandgap, so well below the *BCD_A_*^+^ donor level located at 0.286 eV from the conduction band. However, in our experiments, the electrons are injected to the sample by a small forward current of only 5.7 µA. The quasi-Fermi level of the non-equilibrium electrons injected in this case cannot rise much above the middle of the bandgap, and so a small pre-exponential factor is expected in our case too. 

For the reverse reaction, switching from B to A configuration (see Figure 11b), an electron must be emitted from the *BCD_B_^0^*: (17)$${BCD}_{B}^{0}\to {BCD}_{A}^{+}+e$$

The rate of this transformation was determined to be:(18)$$k={1.2\times 10}^{12}\exp\left(-\frac{0.94\;\mathrm{eV}}{k_{B}T}\right)$$

The pre-exponential factor, in the order of 10^12^ s^−1^, is in the expected range for a free carrier emission process [59,60]. The large barrier of 0.94 eV for this transformation determines the long-time variations observed in *N_eff_* at ambient temperatures once the sample receives an excess of electrons, even by a short exposure of similar high-resistivity samples to ambient laboratory light [32]. On the other hand, such variations in *C-V* characteristics are not observed in low-resistivity diodes of 10 and 50 Ωcm [39,40]. Compared to low-resistivity *p-type* samples, in high-resistivity ones, the number of electrons induced by a small forward current, comparable with the photocurrent generated by ambient light, is larger than that of injected holes and on longer distances inside the diodes. Therefore, the change in the BCD configuration from A to B is expected to take place in a much smaller volume in low-resistivity samples than in the high-resistivity ones, causing, most likely, negligible variations in the *C-V* characteristics. Further experiments are planned to be conducted to investigate the bistable behavior of BCD defects in samples with resistivity of 1 kΩcm, 250 Ωcm, and 100 Ωcm. 

## 5. Conclusions

The present study was conducted due to significant changes being observed in the *C-V/I-V* characteristics of PAD and LGAD structures, fabricated on high-resistivity *p-type* silicon wafers, caused by their inevitable exposure to ambient light prior to electrical measurements. The values of *N_eff_* determined from such characteristics vary as well, making it thus difficult to understand radiation damage and to develop reliable models for the acceptor removal process in *p-type* silicon. The reason behind such variations is the bistable behavior of BCD defect, a boron-containing radiation-induced complex detected in the present study by means of TSC experiments.

The experiments performed in this work allow us to reveal the kinetics behind the BCD defect configurational transformations and to determine the electronic properties of two structural configurations, A and B. Thus, in equilibrium conditions, the BCD defect is found in the A configuration. From TSC investigations, the corresponding zero-field energy level for the *BCD_A_*^(0/+)^ donor state was determined to be located at *E_t,_*_0_ = 0.286 eV from the conduction band of silicon. In non-equilibrium conditions, the defect can capture a free electron at ambient temperatures, overpassing the energy barrier of *E_A__→B_* = 0.363 eV needed for changing into the neutral-charged B configuration (*BCD_B_*^0^). The defect remains in this *BCD_B_*^0^ state as long as electrons in excess exist in the sample. Removing the source of non-equilibrium carriers, the defect returns back in its donor configuration *BCD_A_*^(0/+)^ by emitting an electron which surmounts the *E_B__→A_* = 0.94 eV large barrier. Although the present study does not elucidate the microscopic atomic structure of the BCD defects, it reveals the physical mechanisms behind the conversions between two defect configurations and the consequences these transformations have on the electrical characteristics of the device. 

## Figures and Tables

**Figure 1 sensors-23-05725-f001:**
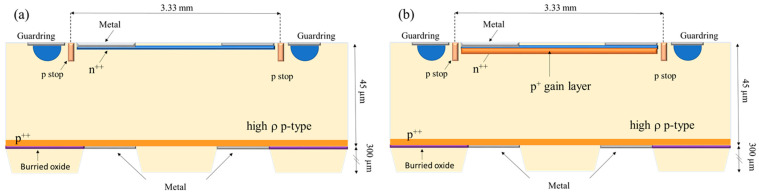
Schematic transversal cross-section of the investigated samples: (**a**) PAD diode; (**b**) LGAD.

**Figure 2 sensors-23-05725-f002:**
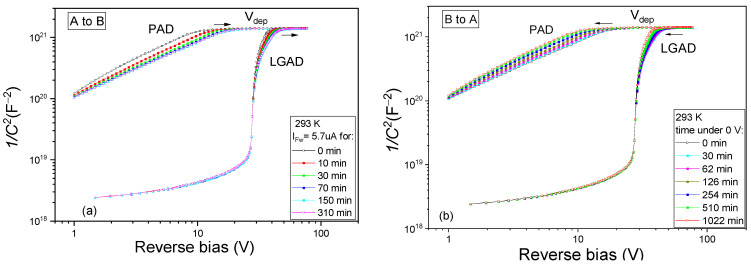
*C-V* characteristics of irradiated PAD and LGAD samples recorded at 293 K for: (**a**) different injection times with I_Fw_ = 5.7 µA when the BCD defect changes from A to B configuration; (**b**) different times passed from the end of injection, with samples kept in dark and under 0 V, when the BCD defect returns back in the A configuration.

**Figure 3 sensors-23-05725-f003:**
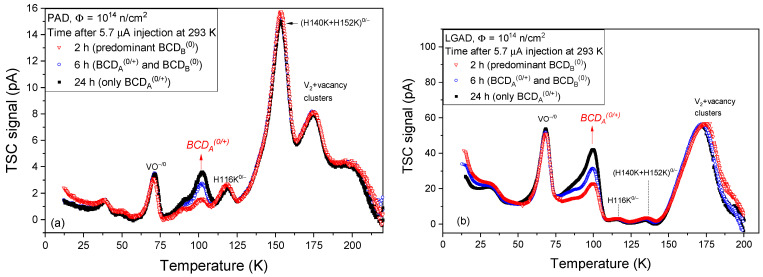
TSC spectra, recorded at different time intervals after the switch of BCD defect from A to B configuration, were achieved with 3 h of 5.7 µA forward current injection at 293 K on: (**a**) PAD with a reverse bias of V_R_ = 100 V; (**b**) LGAD diodes with V_R_ = 200 V. The TSC measurements after 24 h (the black curves) reveal the maximum amount of *BCD_A_*^(0/+)^ defects detected in our experiments.

**Figure 4 sensors-23-05725-f004:**
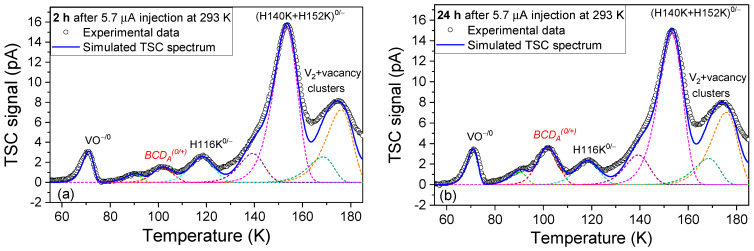
Simulation of the TSC signal measured with V_R_ = 100 V at different time intervals after the BCD defect switched from A to B configuration in the PAD diode: (**a**) 2 h; (**b**) 24 h. The colored dashed lines represent the simulated TSC signals corresponding to each of the defects detected in this study and the straight blue line is the resulting total simulated TSC spectrum.

**Figure 5 sensors-23-05725-f005:**
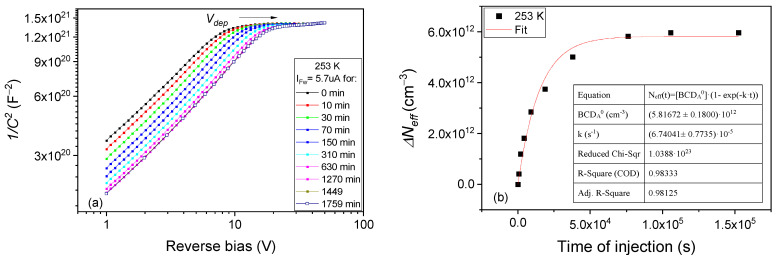
PAD diode injection with I_Fw_ = 5.7 µA at 253 K: (**a**) *C-V* characteristics measured after different times of injection; (**b**) variation in *N_eff_* with injection time. Fit according to Equation (13).

**Figure 6 sensors-23-05725-f006:**
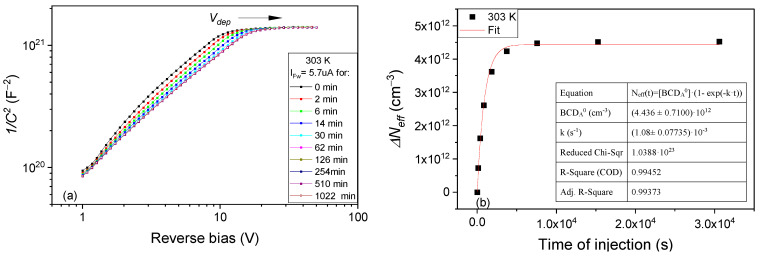
PAD diode injection of with I_Fw_ = 5.7 µA at 303 K: (**a**) *C-V* characteristics measured after different times of injection; (**b**) variation in *N_eff_* with injection time. Fit according to Equation (13).

**Figure 7 sensors-23-05725-f007:**
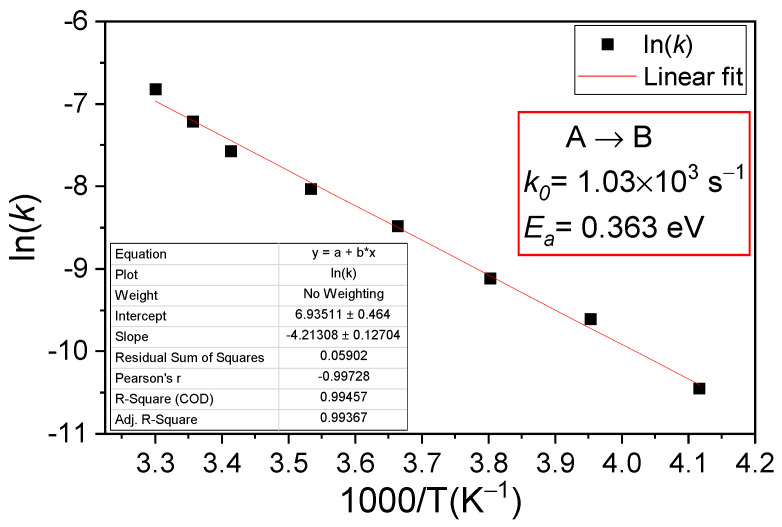
Arrhenius plot of transformation rates determined for temperatures between 243 K and 303 K on PAD diode when the BCD defect changes its configuration from A to B.

**Figure 8 sensors-23-05725-f008:**
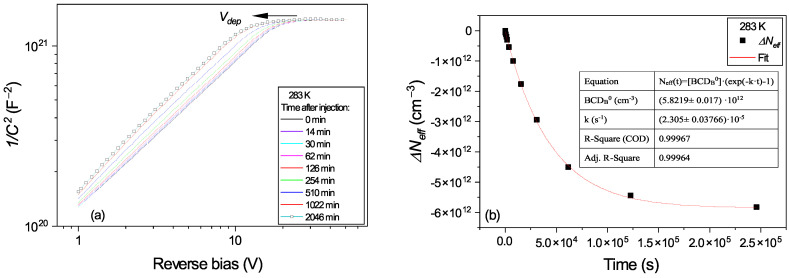
PAD diode after the 5.7 µA forward current injection was performed at 283 K: (**a**) *C-V* characteristics measured at different times after injection; (**b**) variation in *N_eff_* in time. Fit according to Equation (14).

**Figure 9 sensors-23-05725-f009:**
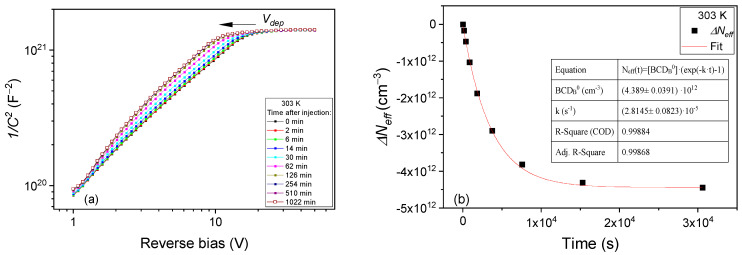
PAD diode after the 5.7 µA forward current injection was performed at 303 K: (**a**) *C-V* characteristics measured at different times after injection; (**b**) variation in *N_eff_* in time. Fit according to Equation (14).

**Figure 10 sensors-23-05725-f010:**
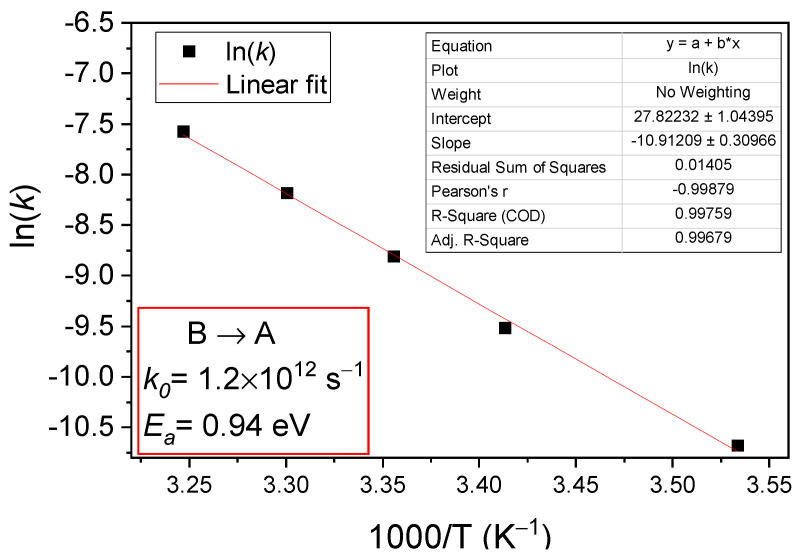
Arrhenius plot of transformation rates determined for temperatures between 283 K and 308 K on PAD diode when the BCD defect returns from B to A configuration.

**Figure 11 sensors-23-05725-f011:**
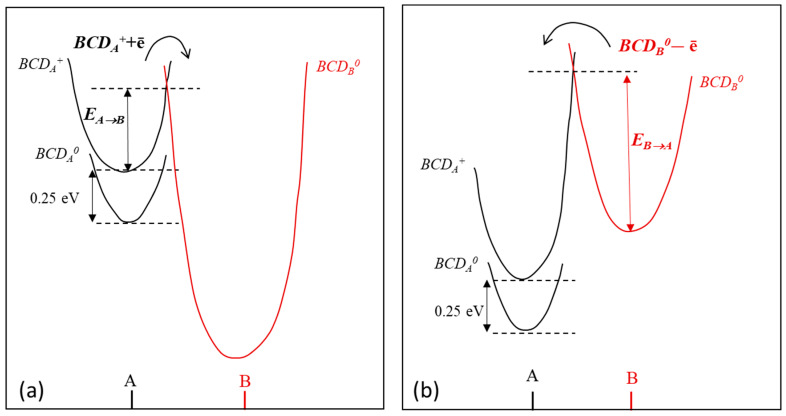
Configuration coordinate diagram for the BCD defects in silicon: (**a**) in non-equilibrium conditions under small forward injection; (**b**) in the absence of free electrons in excess. The configurational coordinate—energy curves are represented in black and red for the A and B configurations, respectively. The arrows indicate the transition points between the defect configurations, accompanied by the capture (**a**) or by the emission of electrons. The experimental values for change in the BCD structural configuration are *E_A__→B_* = 0.363 eV and *E_B__→A_* = 0.94 eV.

## Data Availability

Not applicable.

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
