# Peer review of "Bistable Boron-Related Defect Associated with the Acceptor Removal Process in Irradiated p-Type Silicon—Electronic Properties of Configurational Transformations"

_sensors, 2023, doi:10.3390/s23125725_

Round 1

Reviewer 1 Report

In this work, the authors examined the electronic properties of the Boron Containing Donor. Energy barriers of 0.36 eV and 0.94 eV were claimed to be determined for configuration transformations A to B and B to A, respectively. It shows a steady configuration. Therefore, the authors should provide readers with all information pertaining to the transition configuation, such as the oxidation status of boron species.

Author Response

We thank the reviewer for the time spent in evaluating our work. Below is our response (in itallic) to reviewer comments.

In this work, the authors examined the electronic properties of the Boron Containing Donor. Energy barriers of 0.36 eV and 0.94 eV were claimed to be determined for configuration transformations A to B and B to A, respectively. It shows a steady configuration. Therefore, the authors should provide readers with all information pertaining to the transition configuration, such as the oxidation status of boron species.

Response: Without knowing the structural conformation of the BCD defect and based only on electrical measurements, we can only assess the activation energies and the frequency factors for the defect transformation from A to B and back configurations. Further investigations based on DFT calculations can be pursued on the most probable chemical-structures of the BCD, to determine the oxidation state of boron species and the properties of the transition state, but these are beyond the scope of the present study. However, we corrected the former Eqs. (6) and (8) (Eqs. 15 and 17 in the revised manuscript), describing the A-B and B->A processes, respectively.

Reviewer 2 Report

See the Attachment.

Minor editing of English language is required.

Author Response

We are thankful for the reviewer effort to read our manuscript. Below is our response (in italic) to reviewer comments.

Silicon detectors of radiation are widely used in different applications. The most demanding one is their operation in the high-intensity radiation fields required for fundamental research in high-energy particle physics experiments. The impact of such particles on silicon displaces the atoms from their sites and forms electrically active defects able to degrade the detector performance and limit its practical use.
At high irradiation fluences, the main problem for n-Si silicon detectors is the conduction type inversion determined by the radiation induced point and clustered acceptor-like defects leading to losses in their spatial resolution. A solution proposed was to develop the p-Si based detectors. But, they reveal an acceptor removal effect: the losing of the initial acceptor doping of the material during its irradiation.

The acceptor removal process is explained by considering a reaction between the substitutional acceptor dopant, usually, boron B_s and the silicon interstitial Si_i created by irradiation. The traditional view accounts for two-steps process: (1) B_s atoms switch the place with the Si_i becoming boron interstitials B_i and then losing their acceptor character; and (2) B_i atoms migrate in the crystal, combine with abundant oxygen (the main background neutral impurity in silicon) interstitials O_i and form the B_iO_i complex defects. The more recent mechanism accounts for the reaction between negatively charged acceptor dopant B_s and the positively charged Si_i created by
irradiation. There is formed a B_s–Si_i complex defect with different charge states and electronic configurations. Each of these two boron-containing complex defects, B_iO_i and B_s–Si_i, has in silicon a donor energy level (and, thus, in corresponding charge state contributes twice its concentration to the reduction of the initial p-type doping) and is referred as Boron Containing Donor (BCD). The radiation induced BCD is a bistable defect being able to reversible switch between two structural configurations:
electrically active donor state with energy level of 0.25 eV and neutral state. The change between two configurations can be triggered by an excess of carriers at ambient temperatures achieved.

Paper aims to determine the BCD’s properties in its different electronic configurations, as well as the configurational conversions kinetics in usual laboratory conditions.
These properties and the desired kinetics behind configurational transformations are determined from the variations in the capacitance–voltage characteristics in the 243 – 308 K temperature range. Energy barriers of 0.36 and 0.94 eV are estimaated, respectively, for transformations from electrically active donor state to neutral one and back.
As there is no clear identification of the BCD defect and the physical origin of its bistability, Authors discuss the phenomenology behind the BCD conversion between its configurations with the help of a configuration-coordinate diagram based on the determined energy barriers in their experimental study.
â–  We recommend adding the Paper with references to some works by Pagava et al.:

 – Refs. to reports on conduction type inversion in irradiated n-Si (Lines 45-49) should be added with note that n–p inversion in silicon can be induced not only by irradiation, but plastic deformation as well [Eureka: Phys. Eng., 2019, 4, 76-81].

Done – the suggested paper is included as reference [9] in the revised manuscript.

– Section “4. Discussion” should be added with analysis of Refs. to reports [Radiat. Eff. Def. Solids, 2006, 161, 12, 709-713; Ukr. J. Phys., 2007, 52, 12, 1162-1164; Ukr. J. Phys., 2012, 57, 5, 525-530; Eur. Chem. Bull., 2013, 2, 10, 785-793; East.-Eur. J. Ent. Technol., 2015, 4, 5 (76) – Appl. Phys. Mater. Sci., 52-58; Nano Res. Appl., 2017, 3, 10 (1-8)] on boron-containing secondary radiation defects in silicon obtained by means of electrophysical measurements.

Among the mentioned papers of Pagava we found one relevant for our study concerning the formation and annealing of BV2 complex in irradiated samples. We included this paper as a new reference ([46] in the revised manuscript) in a discussion, added in the Introduction, about vacancy and interstitial boron related radiation induced defects.

Reviewer 3 Report

The manuscript reports on the investigation of defect reactions in neutron-irradiated specifically designed structures. Presented results are original and contribute to the studies of defects in silicon. The manuscript should be published in Sensors after authors improve the manuscript according notes below. 

1. All presented results are for n-irradiated samples. Authors should inform, if they made the same measurements also on non-irradiated samples and if the comparison of defect structure is in agreement with their model.

2. Values with huge exponents should be rescaled to more appropriate units. For example, current in (A)-Fig.3 can be given in (pA), etc.

3. Defect reactions specified in the Introduction occur in parallel with the formation of Si vacancy V_Si created by irradiation. Authors should discuss possible effect of V_Si on their measurement and argue, why they neglect their effect from the reasoning.

4. It is known that the determination of the defect density from TSC is not safe. Authors should report on the way, how they determined the density 3.8x10^12 cm^-3 presented in line 208.

5. In equation (6), BCD_B^0 should stand on the right hand side.

There are many errors in grammar. Some of them are listed below.

line 74: 'atoms migrates'

line 76: 'all type'

line 86: 'literature both type' -> 'literature, both types'

line 88: 'state contribute' -> 'state, contribute'

line 91: 'device’'

line 106: 'proper' -> 'properly'

line 173: 'This way the' -> 'This way, the'

line 270: 'free electrons the BCD' -> 'free electrons, the BCD'

Round 2

Reviewer 1 Report

na